# Types of anomalies in two-dimensional video-based gait analysis in uncontrolled environments

Yuki Sugiyama [1], Kohei Uno [2], Yusuke Matsui [2,3] *

1 Division of Physical and Occupational Therapy, Department of Integrated Health Science, Graduate School of Medicine, Nagoya University Daiko-minami, Higashi-ku, Nagoya, Japan, 2 Biomedical and Health Informatics Unit, Department of Integrated Health Science, Graduate School of Medicine, Nagoya University Daiko-minami, Higashi-ku, Nagoya, Japan, 3 Institute for Glyco-core Research (iGCORE), Nagoya University, Furo-cho, Chikusa-ku, Nagoya, Japan

☯ These authors contributed equally to this work.
* matsui@met.nagoya-u.ac.jp

**Data Availability Statement:** The relevant data and code used in this study are available on github (https://github.com/matsui-lab/PoseFixeR).

**Funding:** This work was supported by JSPS KAKENHI Grant Number JP20K20657(YM). The

## Abstract

Two-dimensional video-based pose estimation is a technique that can be used to estimate human skeletal coordinates from video data alone. It is also being applied to gait analysis and in particularly, due to its simplicity of measurement, it has the potential to be applied to gait analysis of large populations. However, it is considered difficult to completely homogenize the environment and settings during the measurement of large populations. Therefore, it is necessary to appropriately deal with technical errors that are not related to the biological factors of interest. In this study, by analyzing a large cohort database, we have identified four major types of anomalies that occur during gait analysis using OpenPose in uncontrolled environments: anatomical, biomechanical, and physical anomalies and errors due to estimation. We have also developed a workflow for identifying and correcting these anomalies and confirmed that this workflow is reproducible through simulation experiments. Our results will help obtain a comprehensive understanding of the anomalies to be addressed during pre-processing for 2D video-based gait analysis of large populations.

## Author summary

Gait is one of the important biomarkers of numerous health conditions. With developing mobile health technologies, it is becoming easier to measure our health. However, establishing evidence is a critical issue to providing preventive medicine, we need to be able to collect data from a large population. Two-dimensional video-based pose estimation can be a solution for the gait analysis of such a population. However, the technical accuracy and limitations of this analysis method have not yet been sufficiently discussed. In this study, by analyzing the largest database currently available, we systematically identified four types of technical anomalies that occur during gait measurement: anatomical, biomechanical, and physical anomalies and errors dues to estimation. We have also shown how to deal with these issues and made solutions available as software so that researchers

funders had no role in study design, data collection and analysis, decision to publish, or preparation of the manuscript.

**Competing interests:** The authors have declared that no competing interests exist.

can reproduce them. In the future, increasing numbers of studies will use 2D video-based pose estimation to research health-related gait among large populations. We believe that our work will provide a guideline for researchers and clinicians involved in these studies to discuss design and algorithms.

This is a *PLOS Computational Biology* Methods paper.

## Introduction

Gait is a simple biomarker of the human condition [1], and its effectiveness as a clinical or pre-clinical marker for diseases such as nervous system abnormalities and skeletal muscle abnormalities has been revealed in various fields [2–6]. In recent years, with the advancement of artificial intelligence applications, several gait analysis methods based on computer vision have been proposed [7–10]. These methods are characterized by the extraction of parameters using images or videos of walking as the input. Two such approaches have been proposed to date: one approach is to extract features based on appearance, such as walking silhouette [11, 12], and the other is to extract gait parameters, such as a series of joint positions and joint angles, by fitting human joint models to the images using estimation [13].

One algorithm using the latter approach, OpenPose, can estimate joint coordinates at up to 135 key points, such as "body", "feet", "hands" and "face" for multiple subjects in an image by learning a vector space called Part Affinity Fields (PAF) for associations between anatomical joints based on a deep learning model. Previous research has suggested that this joint estimation capability is sufficient to some extent even in videos with many dynamic factors [14]. Compared with conventional optical motion capture, approximately 80% of the estimated joint coordinates are less than 30 mm with good accuracy [15].

These computer vision-based gait analysis methods can automatically analyze a large number of joint coordinates with only digital video as input and can be used in any environment, including homes and clinics, requiring little time, cost, and effort compared with conventional optical motion capture. Large-scale human gait analysis can be conducted more easily than ever before. However, there are some issues to be solved in gait analysis applications, such as a certain amount of unexpected noise and the false detection of multiple persons even though only one person is walking [16]. In addition, a reproducible and standardized analysis workflow is still lacking. Stenum et al. proposed a comprehensive analysis for obtaining gait parameters based on OpenPose during gait in a controlled environment [16]. This workflow uses video as input, preprocessing of joint coordinates obtained from OpenPose, and extraction of gait parameters such as step length.

However, workflows for gait analysis in uncontrolled environments have not been studied sufficiently. To capture gait futures in large populations in heterogenous environments, a robust approach is needed. Various factors are assumed to potentially affect the accuracy of joint estimation using OpenPose, including camera performance, the distance between the camera and the subject, walking speed, clothing, and walking environment. Several existing studies seem to provide a solution, although they assume a somewhat controlled data acquisition environment. Seethapathi et al. reviewed six categories of pose estimation problems that can hinder the estimation of kinematic parameters in the application of OpenPose in motion science, and suggested several possible solutions; for example, post-processing and elaboration during data acquisition, e.g., size estimation incorporating reference objects [17]. The

workflow reported by Stenum et al. [16] pointed out false person detection and left-right swapping of lower limbs, but it is manual, with detection, correction, and exclusion being based on visual inspection. In an environment that can be controlled to some extent or that includes few subjects, it may be possible to devise data measurement methods or to deal with measurement errors through manual labor. However, both approaches may be limited when measuring large populations of thousands to tens of thousands of people in several different environments, such as in hospitals and other facilities.

When considering the efficient post-processing approach in such an environment, it is useful to perform a statistical examination of the error structure based on large-scale data. Fortunately, in recent years, a gait database consisting of 10,307 individuals has been made public using OpenPose technology [18]. This database contains pose estimations for an unspecified number of visitors to a certain facility during walking at 25 frames per second (fps) using a multi-view camera in an uncontrolled manner, and 18 joints are estimated for one gait cycle [18]. Although this database was originally intended for the biometrics field, we thought technical anomalous errors in uncontrolled environments could be investigated using this data.

The main purpose of this study was to classify the types of anomalies in pose estimation using OpenPose during the gait cycle in an uncontrolled environment to obtain a roadmap for analyzing large-scale gait data with OpenPose. Through our analysis, we identified four main types of anomalies: anatomical, biomechanical, and physical anomalies and errors due to estimation. In addition, we present a data processing workflow for dealing with the errors that we have categorized and demonstrate its reproducibility through simulation experiments. The code used in this study is online (**URL: https://github.com/matsui-lab/PoseFixeR**).

## Results

### Overview of anomaly types

This section provides an overview of the types of anomaly errors in OpenPose measurements during gait that are presented in this paper. Individual anomaly types are discussed in detail in the section below. We conducted a comprehensive analysis of the database (see Materials and Methods), while partially referring to the existing literature [16, 17], and identified four main types of anomalous errors that should be preprocessed when estimating joint coordinates during gait using OpenPose in an uncontrolled environment (Fig 1, Table 1). For convenience, the four types are categorized as anatomical, biomechanical, and physical, as well as errors due to the inherent estimation accuracy of OpenPose. Note that the categories in this study were labeled based on the patterns observed in a data-driven manner, and are used for the convenience of interpretation and to facilitate discussion. Therefore, they are not based on strict anatomical, biomechanical, or physical definitions and these types are not completely independent, as they overlap with each other and, in some cases, are composite. We further subdivided these four categories in terms of detection and correction methods and finally classified them into ten types that we believe should be considered during analysis (Fig 1).

### Anatomical constraints

Anatomical constraints refer to a series of anomalous errors that could be considered deviations with respect to standard human anatomical constraints (Fig 1A). The most common case observed here was extreme lengthening at the joint located on the opposite side to the camera direction (Table 1). This is thought to be caused by forcibly making predictions on unobserved joints. We were unable to find any existing studies explicitly addressing and discussing this

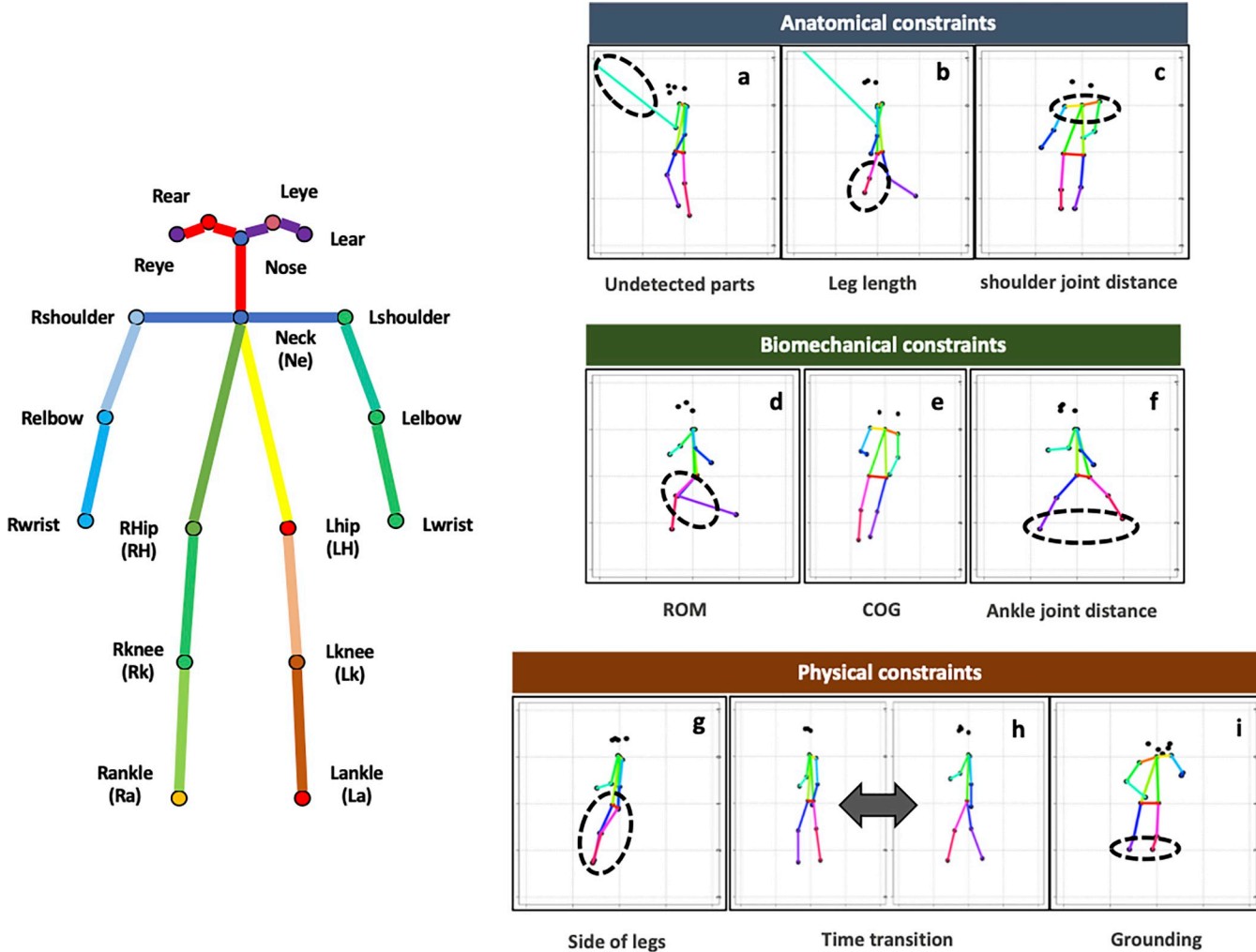

**Fig 1. Types of anomalies during gait using OpenPose.** The left panel shows the OpenPose skeletal model and the name of each part. The right panel shows the anomaly types corresponding to those in Table 1. Estimation accuracy in Table 1 is excluded from the figure for convenience of illustration. ROM, range of motion; COG, center of gravity.

anomaly. However, in approximately 15% of the subjects, we also observed cases where the shoulder width suddenly increased even though the video was taken from the side (Figs 1C and S3) and cases where the skeletal length of the lower limbs became extremely short or long before or after a certain point in time, regardless of the camera direction (Fig 1B).

Ideally, we would like to be able to compare the estimated values with the baseline skeletal structure of each person; however, if this is difficult, it may be possible to estimate them by assuming a standard human anatomical skeletal structure and detecting deviations from it. We believe that the relative proportions of standard human skeletal length estimated by a cohort study [19] could be used to predict the joint coordinate values in advance. Specifically, if we consider a normalized coordinate with Neck (Ne) as the origin and apply the standard skeletal length (see Materials and Methods), we can predict the range within which the joint coordinate values should lie. We used this method to identify joint coordinates that were far outside the normal range (see Materials and Methods).

**Table 1. Percentage of each anomaly type.**

| | ID | Anomaly types | Whole body (18 points) | Only lower limbs (6 points) |
|---|---|---|---|---|
| Anatomical constraints | a | Undetected parts | 97.7 | 20.5 |
| | b | Leg length | 14.8 | |
| | c | Shoulder joint distance | 16.1 | - |
| Biomechanical constraints | d | Ankle joint distance | 11.6 | |
| | e | Range of motion | 69.4 | |
| | f | Center of gravity | 6.3 | |
| Physical constraints | g | Side of legs | 29.0 | |
| | h | Time transition | 99.7 | 93.4 |
| | i | Grounding | 2.1 | |
| Estimation accuracy | j | Reliability | 100 | 43.2 |

Anomalies for the whole body are reported as a percentage of the 18 whole body parts, and anomalies for the lower limbs are reported as a percentage of the six lower limbs parts. IDs correspond to the right panel in Fig 1.

## Biomechanical constraints

Anomalous errors deviating from the biomechanical constraints were also observed, mainly in key parameters in the gait analysis, such as the range of motion of the joints (ROM), which is the external angle of the axis connecting Ne and Right Hip (Rh) (or Left Hip [Lh]) and Right Knee (Rk) (or Left Knee [Lk]) (Fig 1D); center of gravity of the trunk (COG) representing the inclination of the trunk (Fig 1E); and stride length with ankle distances (Fig 1F). In particular, the anomalies related to ROM were the highest, accounting for nearly 70% of all subjects (Table 1), demonstrating the difficulty of biomechanical analysis in uncontrolled environments. It should also be noted that the skeletal model in OpenPose does not exactly match actual anatomical skeletal structures (see Discussion).

Furthermore, there are biases in the estimates depending on the distance between the camera and the subject, as well as errors in the pose estimation itself. In fact, when compared with the ROM estimated using a gyro sensor [20], a shift of approximately 10° to 20° was observed, and the variance tended to be large (Fig 2, Table 2). Therefore, instead of directly applying criteria based on other measurement methods, such as gyroscope sensors, OpenPose's baseline should be estimated to separate the signal from the noise. We detected unnatural errors biomechanically, by calculating thresholds based on statistical confidence intervals derived from the database (see Materials and Methods).

Although COG and stride length anomalies were relatively infrequent, they were distinctly different from the COG and stride length of natural gait and, thus, had to be detected and corrected. In a healthy person, COG is unlikely to fluctuate significantly throughout the gait cycle. We used clustering to identify the positions of the hip (Rh or Lh), knee (Rk or Lk), and ankle (Right Ankle [Ra] or Left Ankle [La]), which were considered to be off-center (see Materials and Methods, S1 Fig).

For stride length, we observed cases where the distance of the ankle joint (Ra and La) was underestimated or overestimated for a particular frame or for the entire frame. For the other cases, we focused on the maximum stride length in the gait cycle and derived a threshold based on statistical confidence intervals to identify the error (S2 Fig).

## Physical constraints

Gait is a continuous motion in time that depends on the frame rate of the video recording in OpenPose, but it is difficult to imagine instantaneous motion beyond the physical constraints

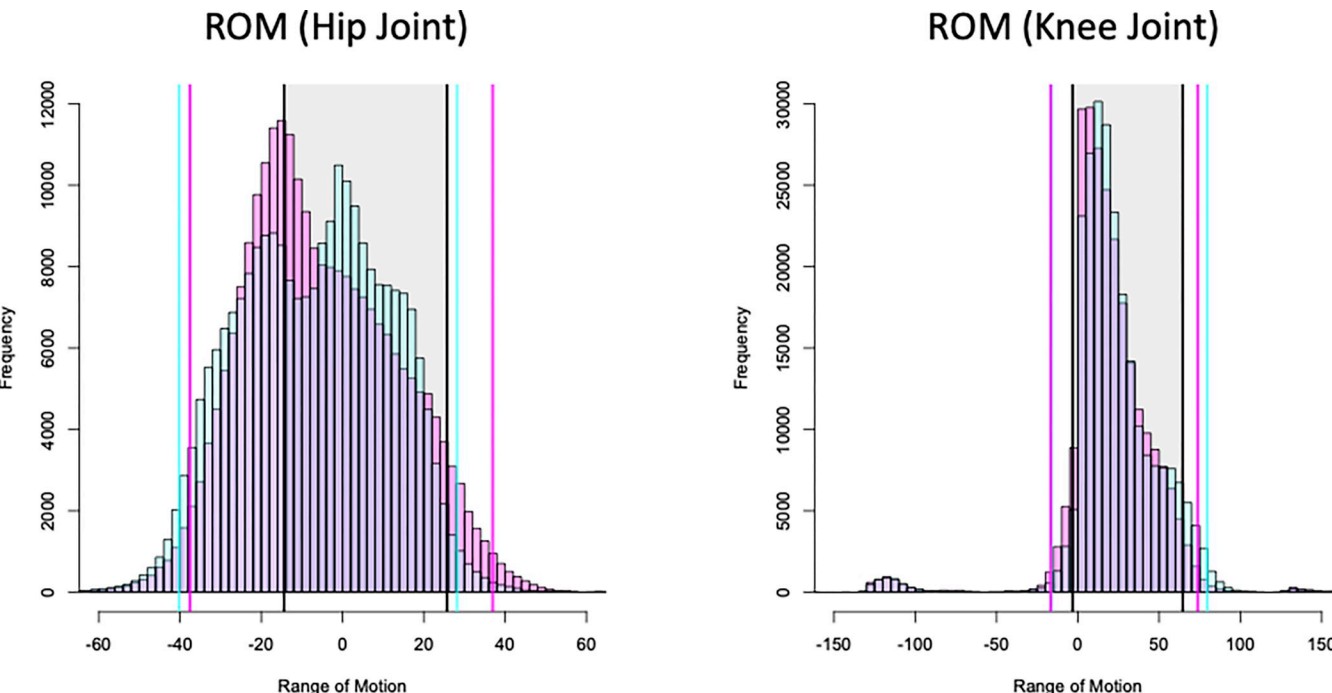

**Fig 2. Histograms of ROM of hip and knee joints (left direction: red, right direction: blue).** The ranges in red and blue shading are 95% confidence intervals. The black shaded area indicates the maximum and minimum ROM during gait using inertial sensors as reported by Park et al. ROM, range of motion [20].

**Table 2. Mean shift and variability of ROM at the hip and knee joints, comparing published gyro-sensor-based statistics for ROM with those obtained in the present database analysis.**

| (Unit:deg) | Variable | System | Mean±SD (before anomaly exclusion) | Mean±SD (after anomaly exclusion) |
|---|---|---|---|---|
| Hip-joint angle (+ flexion/—extension) | Max | OpenPose (R) | 27.00±12.40 | 23.88±8.08 |
| | | OpenPose (L) | 22.28±9.81 | 17.27±6.12 |
| | | MocapNET | 25.70±3.85 | 25.70±3.85 |
| | Min | OpenPose (R) | -33.07±10.50 | -28.35±6.08 |
| | | OpenPose (L) | -36.37±11.17 | -32.14±5.94 |
| | | MocapNET | -14.41±2.23 | -14.41±2.23 |
| | ROM | Open Pose(R) | 60.06±14.86 | 52.23±8.96 |
| | | Open Pose(L) | 58.65±14.60 | 49.41±7.74 |
| | | MocapNET | 39.88±3.22 | 39.88±3.22 |
| Knee-joint angle (+ flexion/—extension) | Max | Open Pose(R) | 64.20±21.83 | 58.17±9.63 |
| | | Open Pose(L) | 71.29±20.88 | 64.43±10.87 |
| | | MocapNET | 64.58 ± 5.21 | 64.58 ± 5.21 |
| | Min | OpenPose (R) | -4.88±8.29 | -1.54±6.24 |
| | | OpenPose (L) | -1.10±7.05 | 0.28±6.26 |
| | | MocapNET | -3.18±3.11 | -3.18±3.11 |
| | ROM | OpenPose (R) | 68.34±19.74 | 59.71±11.08 |
| | | OpenPose (L) | 72.39±18.99 | 64.15±11.98 |
| | | MocapNET | 67.20±4.66 | 67.20±4.66 |

ROM, range of motion; SD, standard deviation

in the normal range. Therefore, motion with extremely discontinuous changes is considered to be due to errors. We considered two types of errors: reversals of the left and right legs (Fig 1G) and discontinuous frame transitions (Fig 1H). In the latter case, it was sufficient to detect the change point in the time series. However, it was not sufficient to detect the point at which the legs switched; thus, the deviated state was detected using the direction vector in periodic motion (see Materials and Methods). Another physically unnatural case was floating above ground surface (Fig 1I). In this case, the normal range from the head to the ankle was estimated in advance based on the standard human skeletal model [19], and deviations from this range were detected (see Materials and Methods).

## Estimation accuracy

The reliability score in OpenPose is calculated based on the distance from the correct location to each pixel in the image [21]. The pixels that are the shortest distance from the correct location are considered to have the highest reliability, whereas a low reliability score suggests that the estimated joint may not exist in the image or that it cannot be estimated. The variability of the actual reliability score tended to be lower for certain joints (Fig 3B). It was not easy to identify the cause of the error based on the reliability score, which is an inherent problem in the deep learning algorithm of OpenPose. However, it is possible to determine the reliability score from a statistical perspective. The overall distribution of the reliability scores was bimodal (Fig 3A), suggesting the existence of two potential groups of low and high reliability. We estimated these two groups based on k-means clustering and detected the group with low accuracy.

## Accuracy of pose estimation during gait in an uncontrolled environment

We used a largescale database of OpenPose gait data to examine the reliability of OpenPose estimates of joint coordinates. However, the database used in this study only contained Open-Pose data, so it was not possible to evaluate the data using external criteria. Instead, using our workflow, we calculated percentages by defining joint coordinates that did not contain the anomalies shown in Table 1 as negative examples of the anomalies. The percentages of each joint determined to be normal and the percentage after correction using our workflow are summarized in Fig 4. The workflow will be described in the next section.

The accuracy for the joints on the opposite side to the camera direction was low, but on the same side, the accuracy varied, ranging from 53.8% to 93.5%. In particular, the knee (Rk or Lk) and ankle (Ra or La) contained some anomalous errors in nearly half of the subjects, strongly suggesting that they need to be addressed for downstream analysis to be performed properly.

## Workflow for anomaly detection and correction

To perform gait analysis using OpenPose in an uncontrolled environment, many anomalous errors must be addressed during preprocessing. However, it is unclear which strategies should be used for detection and correction. Here, we present a workflow for detecting and correcting 10 types of anomalies (Fig 5).

## Normalization step

The first normalization step transforms the coordinate system and skeletal length into a form that is comparable for all samples. Ne was set as the origin and transformed into joint coordinates corresponding to the ratio of the neck to the trunk length (see Materials and Methods). This allows for a general discussion of statistical properties and the setting of thresholds to deal with anomalous errors, which allows for efficient preprocessing.

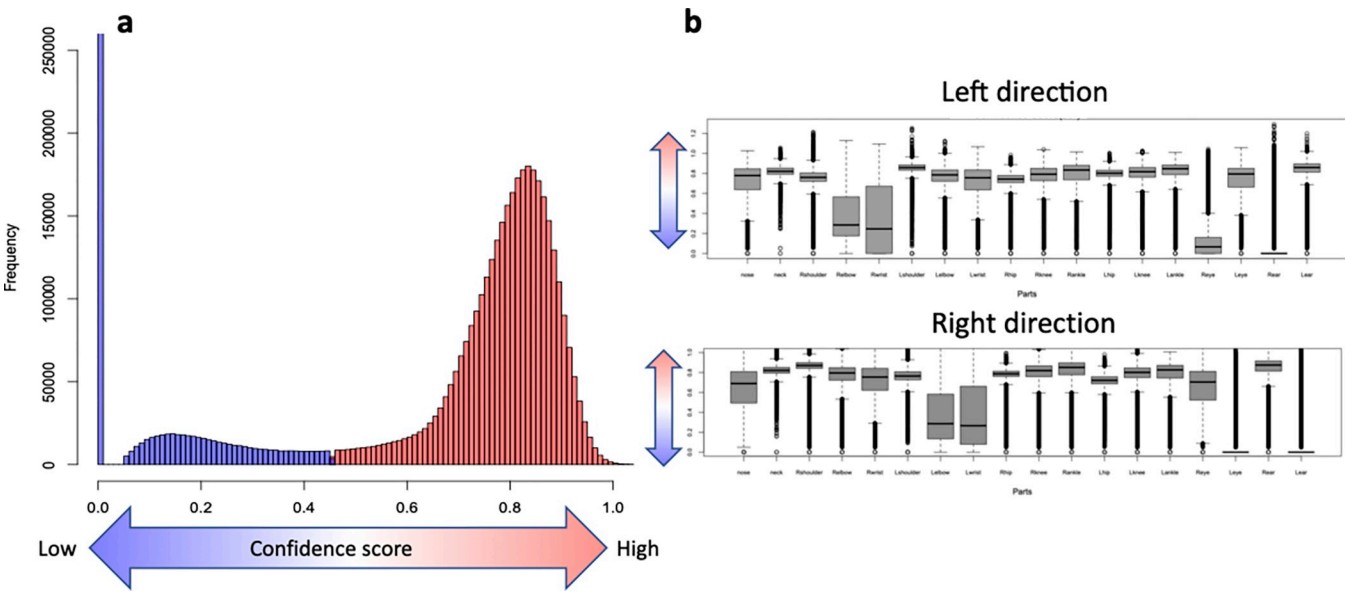

**Fig 3. Reliability score.** (a) Distribution of reliability scores for all subjects. Two groups, low confidence (blue) and high confidence (red), were assumed and classified by clustering. (b) Reliability score per part. The upper panel shows the left direction, and the lower panel shows the right direction.

## Anomaly detection step

Following normalization, the anomalous error of switching the left and right legs was first detected and corrected. This is because leg swapping is a serious error in gait analysis and may affect the detection of other anomaly types. Subsequently, the other nine types were detected.

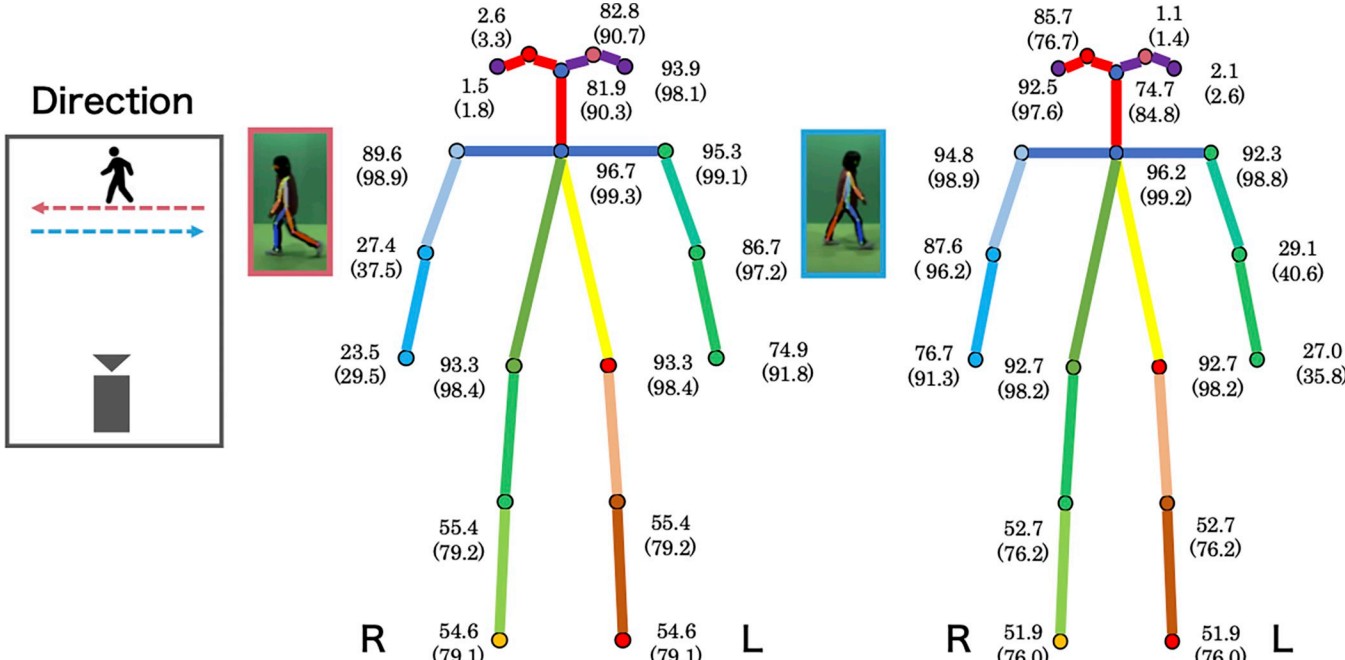

**Fig 4. Estimated accuracy for each part with OpenPose.** The accuracy based on the anomaly types (i.e., the percentage not containing any of the anomalies listed in Table 1) is shown. The numbers in parentheses represent the accuracy after correcting the proposed workflow.

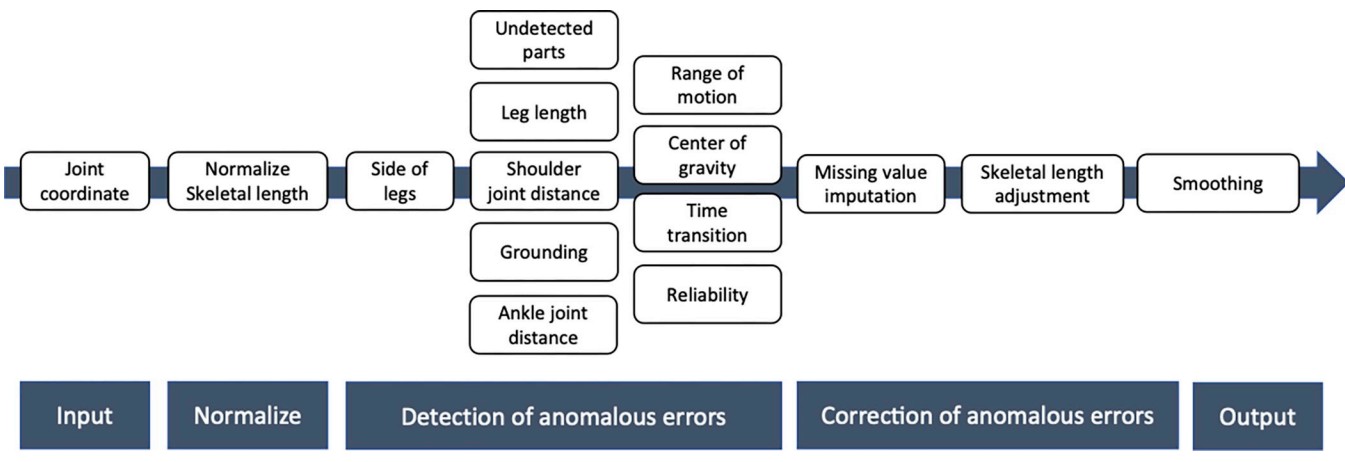

**Fig 5. Proposed workflow.**

## Correction of anomalous error step

The detected parts with anomalous errors could be considered missing values because they cannot be used in the downstream analysis. To some extent, they can be imputed using information from the previous and subsequent frames via averaging. However, if the overall number of missing values is extremely high in one gait cycle or if there are many continuous missing values, the reliability of the missing-value imputation is itself questionable, and such subjects should be excluded. Two filtering criteria were used: (1) the error percentage of each part of the total number of frames was greater than 40%, and (2) the percentage of consecutive missing frames within a gait cycle was more than 20% of the total number of frames (see Materials and Methods). Finally, we were able to impute the missing values of 66.8% of the participants.

## Other adjustments

In addition, because it has been reported that video-based pose estimation causes distortions in the estimated coordinate values depending on the distance from the camera (Fig 6), and because the skeletal length within the same subject is not constant [16], we also corrected the skeletal length (see Materials and Methods).

## Reproducibility of workflow

**Simulation model.** In order to validate the reliability of the proposed workflow, it is common to compare the results of it with ground-truth data such as BICON. However, since there is no ground-truth data used in result section; thus, we conducted a simulation experiment based on actual data (S1 Text) to validate the reproducibility of the workflow in this study. First, we extracted some samples from a real dataset. Second, we generated a pseudo dataset by adding various errors to these samples. The probability of occurrence of each anomalous error was calculated using relative frequencies (Table 1). We generated 10,000 subjects with 25 frames per gait cycle and evaluated the detection accuracy for each anomaly type and the reproducibility of the true joint coordinates.

**Simulation results.** First, we confirmed the reproducibility of the detection accuracy for individual anomalous errors each parts; the sensitivity and specificity were 82.6% and 95.1%, respectively. However, the accuracy for each type of anomaly varied from 71.1% to 95.4%,

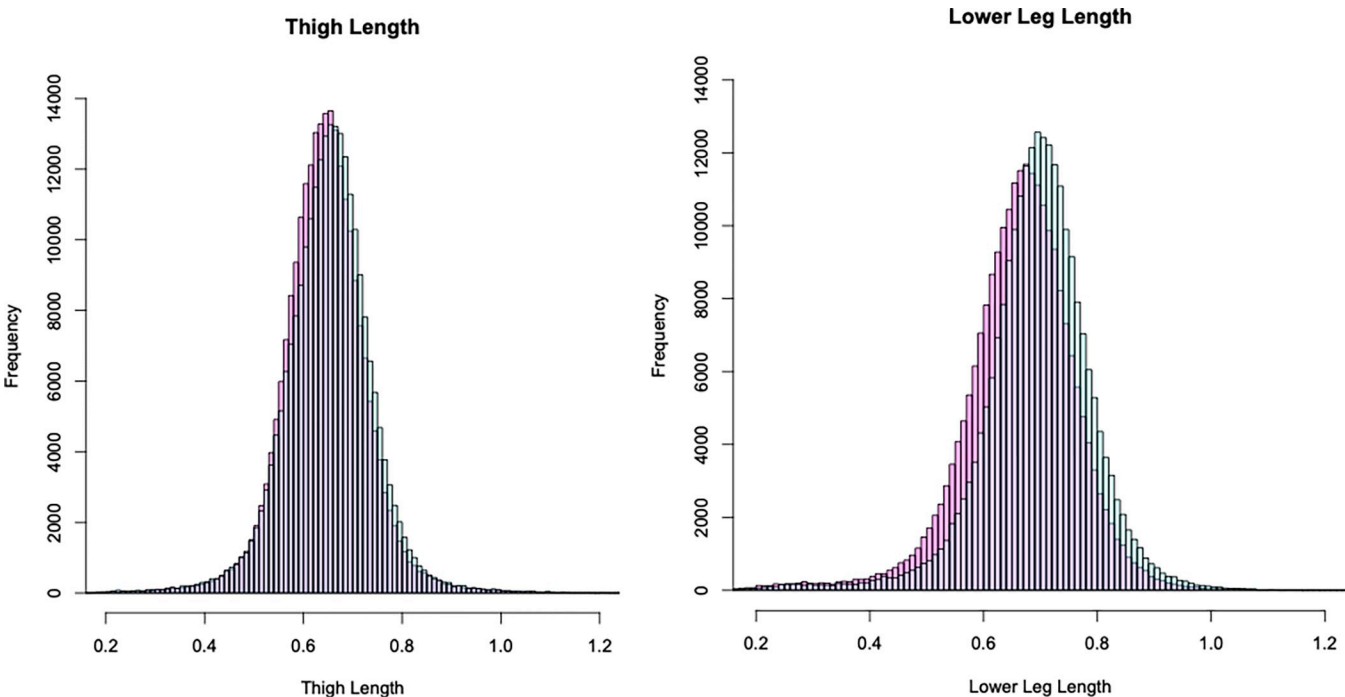

**Fig 6. Skeletal length of legs.** The left and right panels show the skeletal lengths of the thigh and lower leg, respectively, which depend on the distance between the camera (from the right side) and the subjects. The red and blue represent the right and left sides of the body, respectively.

which indicates that the difficulty of anomaly detection varies from joint to joint (Table 3). Pearson's correlation coefficients of the xy-coordinate values before and after the correction were calculated to determine the accuracy for missing imputation, suggesting that there was a significant reproducibility for the joints overall (0.770 for the x-coordinate; p-value <2.2e-16, 0.961 for the y-coordinate; p-value <2.2e-16). In addition, the reproducibility of the individual joints was evaluated. Regarding the accuracy at the level of individual joints, there was a tendency for the accuracy to be relatively low for undetected joints, leg length, COG, and ankle distance, which is thought to represent the variability due to the relatively low accuracy of anomaly detection (Table 4).

**Table 3. Sensitivity of detection of each type of anomaly by workflow.**

| Anomaly types | Sensitivity (%) |
|---|---|
| Undetected parts | 73.0 |
| Leg length | 71.1 |
| Shoulder joint distance | 85.2 |
| ROM | 82.4 |
| COG | 79.3 |
| Ankle joint distance | 93.0 |
| Time transition | 95.4 |
| Side of legs | 88.1 |
| Grounding | 89.4 |

ROM, range of motion; COG, center of gravity

**Table 4. Reproducibility of values by workflow.**

|  | Correlation (x-coordinate) | Correlation (y-coordinate) |
| --- | --- | --- |
| Undetected parts | 0.691 | 0.935 |
| Leg length | 0.767 | 0.929 |
| Shoulder joint distance | 0.768 | 0.946 |
| Ankle joint distance | 0.734 | 0.922 |
| ROM | 0.946 | 0.943 |
| COG | 0.814 | 0.988 |
| Ankle joint distance | 0.734 | 0.922 |
| Time transition | 0.888 | 0.952 |
| Side of legs | 0.657 | 0.967 |
| Grounding | 0.937 | 0.871 |

Pearson's correlation coefficient between the true value and the estimated value with correction after performing all the steps of the workflow. ROM, range of motion; COG, center of gravity

## Implementation

The workflow presented in this paper has been deposited in R code (Github **URL: https://github.com/matsui-lab/PoseFixeR**). A series of preprocessing steps were performed using the coordinate values obtained from the posture estimation using OpenPose as the input data. Detailed parameter settings are described in a vignette.

## Discussion

We used a large database analysis to comprehensively classify technical anomalies using OpenPose gait pose estimation and identified four main types: anatomical, biomechanical, and physical anomalies and errors dues to estimation. We have also presented a method of detecting these anomalies and suggested a workflow for their correction. According to our criteria, all of the 18 parts estimated by OpenPose contained some anomalies, suggesting that proper pre-processing is required before extracting gait features. Moreover, simulation experiments showed that the accuracy of anomaly detection and correction varied depending on anomaly type, implying the need to develop appropriate preprocessing methods for each type.

In particular, the nature of the two-dimensional video-based representations makes skeletal length distortions dependent on the distance between the camera and the subject, and produces anomalous joint coordinate estimates owing to unobserved joints on the opposite side to the camera. The latter anomalous measurement error could be rescued with up to 73% accuracy by our workflow, as shown in the numerical experiments, and it was difficult to capture the complete characteristics of one gait cycle by video recording only on one side. Therefore, it may be effective to develop an experimental design that focuses on specific parameters, such as the motions of specific joints on the video recording side.

The anatomical skeletal models should also be evaluated. In the comparison of inertial sensors and OpenPose in terms of ROMs, we observed a shift of approximately 10˚–20˚. One of the main reasons for this is the difference in the skeletal model. Taking the hip joint as an example, in the field of orthopedics and rehabilitation, ROM is generally measured by measuring the external angle composed of the axis between the trunk and femur. However, because the hip joint area is simplified in OpenPose, the ROM is calculated from the external angle composed of the axis directly connecting Ne and Rh/Lh, Rh/Lh, and Rk/Lk, which results in different criteria and generates a bias. Therefore, when analyzing gait using OpenPose,

especially when interpreting biomechanical features, we should not simply compare the results with standard sensors, such as gyroscopes and goniometers, but should make comparisons based on OpenPose's baseline.

We showed the anomalies in an uncontrolled environment as comprehensively as possible along with the workflow. However, there are several limitations. First, we did not necessarily cover all anomalies since the results are based on the analysis of a single database. Second, we didn't compare our results to ground-truth data (e.g., motion capture) because only the Open-Pose data was available in the public database. Our proposed workflow should be evaluated in further studies. Third, we determined threshold values (e.g., using in Eq(4)) based on a standard skeletal model [19], so gender and age groups were not considered. For more sophisticated algorithms, appropriate skeletal models should be used. Fourth, there is a problem with OpenPose itself. As Seethapathi et al. [17] pointed out, current pose tracking algorithms do not prioritize measurement of the quantities that are important in movement science, such as three-dimensional position, velocity, and acceleration. Therefore, a new algorithm should be developed that considers important factors in kinematics, so that a more suitable system can be constructed.

In order to increase the clinical applications of video-based 2D pose estimation technology, it will be necessary to find situations where it can be used most effectively from a clinical point of view, and to develop an appropriate analysis algorithm in the future [22]. From a practical standpoint, a reproducible and robust analysis method will be crucial. For example, the development of an algorithm that effectively exploits the latent time-series structure of tracking errors for skeletal coordinates is an important issue for future research. On the other hand, this study also employed some threshold-based methods, such as standard skeletal length ratios derived from existing anatomical knowledge and confidence score thresholds based on statistical distributions. Since the OpenPose skeletal estimates do not take into account anatomical or clinical knowledge, methods that combine existing knowledge may be useful. For example, in the category of "side of legs" (g in Fig 1), which occurs when the left and right legs cross each other, self-evaluation of OpenPose such as frame by frame analysis or coordinate distance between frames cannot detect this anomaly, although it can affect joint angle estimation.

It is also important to study analysis methods for disease signals through clinical research designs, such as comparisons between healthy and diseased groups. For example, there may be an affinity between developing research designs that focus on specific parts of the body and researching algorithms dedicated to the early detection of disease-related signatures. Additionally, the development of open-source software and public databases is also considered to be an important research gap that must be filled to allow further clinical applications and the development of our method as a reproducible research method.

The OpenPose in uncontrolled gait analysis revealed various measurement anomalies in all samples due to technical limitations. However, preprocessing using a combination of anatomical, physical, and biomechanical knowledge and statistical algorithms suggested that nearly 70% of the samples could be rescued, although the accuracy varied for each anomaly type. With the development of appropriate study designs and more sophisticated analysis algorithms in the future, it is expected that accuracy can be improved, even in uncontrolled environments. Since our suggested category (Fig 1) will likely include anomalous errors in the controlled environment, it is considered to be widely applicable, and not limited to pose estimation in unconstrained environments. We hope that our study will be helpful when further studies on large populations have been conducted to accumulate evidence.

## Materials and methods

### Dataset

In this study, we used the Osaka University-Institute of Scientific and Industrial Research (OU-ISIR) Gait Database [18], a multi-view large population dataset with pose sequences [18] deposited in the OU-ISIR Biometric Database. By capturing the subjects walking approximately 10 m back and forth, we can observe the gait cycle at a normal speed for each sample. The images had a frame rate of 25 fps and an image size of $1280 \times 980$ pixels. OpenPose shows the x and y coordinates and confidence levels for 18 joints in each frame: nose, neck, right shoulder, right elbow, right wrist, left shoulder, left elbow, left wrist, right groin, right knee, right foot, left groin, left knee, left foot, right eye, left eye, right ear, and left ear.

### Workflow details

The details of the workflow are described below. First, we define the mathematical notation. Let $(X_{it}[S], Y_{it}[S])$ be the two-dimensional coordinate value of the joint S of subject $i$: $i = 1,2,\ldots N$, at time $t$: $t = 1,2,\ldots,T$. To represent an arbitrary joint, the name of the joint is written with a dot symbol, as in $(X_{it}[\cdot], Y_{it}[\cdot])$.

### Normalization

In video-based pose estimation, normalization is necessary because the skeletal length varies depending on the height of the subject, the distance from the subject, and the position of the camera. In this study, we followed the method described by An W et al. [18] and normalized the skeletal coordinate values in three steps: (1) centering with the Ne coordinates as the origin, (2) estimating the scale factor of the skeletal length, and (3) normalizing the skeletal coordinates. Specifically, centering was performed using Eq (1).

$$(X_{it}^*[\cdot], Y_{it}^*[\cdot]) = (X_{it}[\cdot], Y_{it}[\cdot]) - (X_{it}[Ne], Y_{it}[Ne]) \tag{1}$$

Next, to calculate the "relative scale" for each individual frame, the distance from the midpoint of the post-transformed coordinates of the left hip $(X_{it}^*[LH], Y_{it}^*[LH])$ and the right hip $(X_{it}^*[RH], Y_{it}^*[RH])$ to the neck (0, 0) was calculated as $scale_{it}$.

$$scale_{it} = \frac{1}{2}\sqrt{(X_{it}^*[LH] + X_{it}^*[RH])^2 + (Y_{it}^*[LH] + Y_{it}^*[RH])^2} \tag{2}$$

Finally, we normalized all the joint coordinates such that the relative $scale_{it}$ became 1. That is, each joint coordinate was divided by the relative scale to obtain the normalized coordinates $(X_{it}^\dagger[\cdot], Y_{it}^\dagger[\cdot])$.

$$\left(X_{it}^\dagger[\cdot], Y_{it}^\dagger[\cdot]\right) = \left(X_{it}^*[\cdot], Y_{it}^*[\cdot]\right) \times \left(\frac{1}{scale_{it}}\right) \tag{3}$$

Henceforth, we will use $(X_{it}^\dagger[\cdot], Y_{it}^\dagger[\cdot])$ in the following description.

### Anatomical constraints

According to the skeletal length in the standard skeletal model, if the length of the trunk, which is the distance from the neck to the hip joint, is 1,then the distance from the neck to the top of the head is 0.362, and the distance from the neck to the ankle joint is 2.283 [19]. If the x-coordinates of the top of the head and ankle joint, which are the two ends on the y-axis, are the same as the x-coordinates of the neck, then the neck coordinates are (0, 0), the head

coordinates are (0, -0.362), and the ankle coordinates are (0, 2.283). Therefore, the y-coordinates of all joints were considered to be in the range [-0.362, 2.283]. If we introduce an error ratio (*ER*) to account for individual differences, we can consider that the coordinates of any joint $Y_{it}^\dagger[\cdot]$ lie within the following range:

$$-0.362 \times ER \leq Y_{it}^\dagger[\cdot] \leq 2.283 \times ER \tag{4}$$

Coordinates that do not satisfy this condition deviate from the expected range.

## Biomechanical feature

Here we describe anomaly detection for range of motion (ROM), which is the external angle of the axis connecting Ne and Rh (or Lh) and Rk (or Lk). To identify the joint coordinates deviating from the standard ROM, 95% confidence intervals were constructed based on the empirical distributions derived from the maximum flexion angles of each joint of the lower extremities (both hip and knee joints) obtained by the following calculations, and those outside the confidence intervals were considered anomalies. First, the Rh flexion angle (calculated in the same way as the Lh flexion angle) was calculated as follows:

$$cos_{it}[Rh] = \frac{(X_{it}^\dagger[Ne] - X_{it}^\dagger[Rh])(X_{it}^\dagger[Rk] - X_{it}^\dagger[Rh]) + (Y_{it}^\dagger[Ne] - Y_{it}^\dagger[Rh])(Y_{it}^\dagger[Rk] - Y_{it}^\dagger[Rh])}{\sqrt{(X_{it}^\dagger[Rk] - X_{it}^\dagger[Rh])^2 + (X_{it}^\dagger[Ne] - X_{it}^\dagger[Rh])^2} + \sqrt{(Y_{it}^\dagger[Rk] - Y_{it}^\dagger[Rh])^2 + (Y_{it}^\dagger[Ne] - Y_{it}^\dagger[Rh])^2}} \tag{5}$$

The Rk joint flexion angle (calculated in the same way as the Lk joint flexion angle) was obtained as follows:

$$cos_{it}[Rk] = \frac{(X_{it}^\dagger[Rh] - X_{it}^\dagger[Rk])(X_{it}^\dagger[Ra] - X_{it}^\dagger[Rk]) + (Y_{it}^\dagger[Rh] - Y_{it}^\dagger[Rk])(Y_{it}^\dagger[Ra] - Y_{it}^\dagger[Rk])}{\sqrt{(X_{it}^\dagger[Rh] - X_{it}^\dagger[Rk])^2 + (X_{it}^\dagger[Ra] - X_{it}^\dagger[Rk])^2} + \sqrt{(Y_{it}^\dagger[Ra] - Y_{it}^\dagger[Rk])^2 + (Y_{it}^\dagger[Rh] - Y_{it}^\dagger[Rk])^2}} \tag{6}$$

From these joint angles, the empirical distribution *F* was constructed. The joint angles were set as $R_{it}$, which is defined as arccos($cos_{it}[\cdot]$)$\times 180°/\pi$, and

$$F(r) = \frac{1}{NT} \sum_{i=1}^{N} \sum_{t=1}^{T} I(R_{it} \leq r). \tag{7}$$

The empirical distribution *F* for each joint was considered to represent the range of motion distribution in one gait cycle at the population level, including all subjects. Based on this, we constructed a 95% confidence interval $CI_{95\%}$ for each joint and obtained

$$CI_{95\%} = [F^{-1}(0.025), F^{-1}(0.975)]. $$

Observed values outside the confidence interval were considered errors.

For the center of gravity (COG), we considered a point in three-dimensional space consisting of the midpoints of the x-coordinates of the hip (Rh or Lh), knee (Rk or Lk), and ankle (Ra or La) with respect to the perpendicular line from Ne to the ground and identified the group whose distance from the origin deviated using the k-means method. The number of clusters was determined using the gap statistic [23]. The distance *dist* from the origin Ne to a point can be described as follows:

$$dist = \sqrt{\left(\frac{X_{it}^\dagger[Rh] + X_{it}^\dagger[Lh]}{2}\right)^2 + \left(\frac{X_{it}^\dagger[Rk] + X_{it}^\dagger[Lk]}{2}\right)^2 + \left(\frac{X_{it}^\dagger[Ra] + X_{it}^\dagger[La]}{2}\right)^2} \tag{8}$$

For anomalous errors related to ankle joint distance, we used

$$Length_{ankle} = \sqrt{(X_{it}^{\dagger}[Ra] - X_{it}^{\dagger}[La])^2} \tag{9}$$

to construct empirical distributions, derive 95% confidence intervals, and detect values those outside the intervals as deviating errors. However, because some anomalies in undetected joints may result in extremely large leg lengths, confidence intervals were derived after excluding those errors in advance.

## Physical constraints

Regarding the method used to detect when the left and right legs are reversed, the inversion of the leg joint coordinates at frame $t$ was detected by comparing the leg joint coordinates of the two frames before and after. For this purpose, we detected whether inversion occurs at $3 \leq t \leq T - 2$ frames. As we could use time-series information, we removed the effects of missing values and outliers in advance. We linearly interpolated the knee joint coordinates of the vectors $X_{i\cdot}^{\dagger}[La], X_{i\cdot}^{\dagger}[Ra]$ at each frame and then applied spline smoothing to obtain $Z_{i\cdot}[La], Z_{i\cdot}[Ra]$. If the skeletal coordinates measured in one gait cycle are reversed for the left and right legs, the coordinates should not move like a pendulum but should be biased to either the left or right. Thus, one of the following should be true for the inversion of the left and right legs:

$$\frac{1}{T}\sum_{t=1}^{T} I(Z_{it}[La] - Z_{it}[Ra] > 0) < 0.3 \tag{10}$$

or

$$\frac{1}{T}\sum_{t=1}^{T} I(Z_{it}[La] - Z_{it}[Ra] > 0) > 0.7 \tag{11}$$

Here, $I(A)$ is an indicator function that gives 1 if the condition $A$ is satisfied, and 0 if not. In addition, since it was considered that there is a limit to the movement of the legs during gait,

$$|Z_{it}[La]| < 0.4 \tag{12}$$

was assumed to be satisfied. After satisfying these conditions, the direction of the leg joint movement changes after frame $t$, that is,

$$\text{sign}(Z_{it-2}[La] - Z_{it-1}[La]) = \text{sign}(Z_{it-1}[La] - Z_{it}[La]) = \text{sign}(Z_{it-2}[La] - Z_{it}[La]) \tag{13}$$

and

$$\text{sign}(Z_{it}[La] - Z_{it-1}[La])$$

$$= \text{sign}(Z_{it+1}[La] - Z_{it+2}[La])$$

$$= \text{sign}(Z_{it}[La] - Z_{it+2}[La])$$

$$\neq \text{sign}(Z_{it-2}[La] - Z_{it-1}[La]) \tag{14}$$

are satisfied, and the joint coordinates of the left and right legs are considered to be reversed, where sign($\cdot$) is a sign function.

To detect errors in ground contact, the reference value of the y-axis coordinates of the legs was set to 2.283. When the y-axis coordinates of both legs deviated sufficiently from the reference value, either upward or downward, it was determined that the person was not grounded.

That is

$$Y_{it}[Lh] > 1.2 \times 2.283 \ and \ Y_{it}[Rh] > 1.2 \times 2.283 \tag{15}$$

or

$$Y_{it}[Lh] < 0.8 \times 2.283 \ and \ Y_{it}[Rh] < 0.8 \times 2.283. \tag{16}$$

The error of the frame transition was defined a value as more than a certain distance from the coordinate of frame $t$-1 or frame $t$+1. In other words,

$$\sqrt{(X_{it}^{\dagger}[\cdot] - X_{it-1}^{\dagger}[\cdot])^2 + (Y_{it}^{\dagger}[\cdot] - Y_{it-1}^{\dagger}[\cdot])^2} \geq JUMP$$

or

$$\sqrt{(X_{it}^{\dagger}[\cdot] - X_{it+1}^{\dagger}[\cdot])^2 + (Y_{it}^{\dagger}[\cdot] - Y_{it+1}^{\dagger}[\cdot])^2} \geq JUMP \tag{17}$$

When one or more of the following conditions were satisfied, the joint coordinate was treated as an anomalous error. In this case, JUMP = 0.5 (1/25 s comparison) and 0.7 (2/25 s comparison).

## Selecting subjects for imputation

We excluded subjects with many anomalous error frames that we defined because it would be difficult to extract gait features in the downstream analysis. Exclusion criteria were as follows: (1) the error rate of each region was more than 40% of the total number of frames and (2) the missing values were greater than 20% of the total number of frames in one gait cycle (S5 Fig). The first criterion was set considering that errors could be detected in at least 20% of samples, even in controlled environments. In addition, technical errors caused by other factors may occur in uncontrolled environments. For the second criterion, we considered that the maximum percentage of each phase per gait cycle was approximately 20% [24].

## Supporting information

**S1 Text. Details of the simulation.**
(DOCX)

**S1 Fig. Detection of a COG anomaly.** The left panel shows the coordinate values of (X,Y,Z) = (Ankle, Knee, Hip). The best cluster based on the k-means method using the gap static (right panel) is shown by color coding. Clusters very close to the origin were used as normal measurement samples.
(TIFF)

**S2 Fig. Maximum ankle joint distance.** Maximum ankle joint distance within one gait cycle for each subject. To illustrate the distribution clearly, skeletal length errors due to undetected sites are excluded.
(TIFF)

**S3 Fig. Distribution of shoulder joint distance.** Based on clustering, the four groups were further subdivided into four group each. The group with slightly larger shoulder joint distance (blue, the third group from the left in the histogram) and the group with extremely large shoulder joint distance (red, the fourth group from the left in the histogram) were considered to have abnormal errors.
(TIFF)

**S4 Fig. Accuracy of anomaly correction.** The rate of recovery before and after anomaly correction for each part, with workflow. The top and bottom rows show the accuracy for all joints during walking in the right and left directions, respectively, and the left and right rows show the accuracy before and after correction, respectively.
(TIFF)

**S5 Fig. Number of consecutive anomalous frames.** Histogram of the number of consecutive anomaly frames for all samples is shown. The samples with a number of consecutive anomalous frames over 20% of the total number of frames were excluded.
(TIFF)

## Acknowledgments

We would like to thank Editage [https://www.editage.com] for editing and reviewing this manuscript for English language.

## Author Contributions

**Conceptualization:** Yusuke Matsui.

**Data curation:** Yuki Sugiyama.

**Formal analysis:** Yuki Sugiyama.

**Funding acquisition:** Yusuke Matsui.

**Investigation:** Yuki Sugiyama.

**Methodology:** Yuki Sugiyama, Kohei Uno, Yusuke Matsui.

**Project administration:** Yusuke Matsui.

**Resources:** Yusuke Matsui.

**Software:** Yuki Sugiyama, Kohei Uno.

**Supervision:** Kohei Uno, Yusuke Matsui.

**Validation:** Yuki Sugiyama, Yusuke Matsui.

**Visualization:** Yuki Sugiyama.

**Writing – original draft:** Yuki Sugiyama.

**Writing – review & editing:** Kohei Uno, Yusuke Matsui.

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
