## [Decision Letter · Decision Letter 0]

21 Jul 2022

Dear Matsui,

Thank you very much for submitting your manuscript "Types of anomalies in two-dimensional video-based gait analysis in uncontrolled environments" for consideration at PLOS Computational Biology.

As with all papers reviewed by the journal, your manuscript was reviewed by members of the editorial board and by several independent reviewers. In light of the reviews (below this email), we would like to invite the resubmission of a significantly-revised version that takes into account the reviewers' comments.

We cannot make any decision about publication until we have seen the revised manuscript and your response to the reviewers' comments. Your revised manuscript is also likely to be sent to reviewers for further evaluation.

Sincerely,

Thurmon Lockhart

Guest Editor

PLOS Computational Biology

Feilim Mac Gabhann

Editor-in-Chief

PLOS Computational Biology

Reviewer's Responses to Questions

**Comments to the Authors:**

Reviewer #1: In this manuscript, the authors report on four major types of anomalies that occur during video-based gait analysis using pose estimation and provide a new analysis workflow for correcting these anomalies. Interest in video-based movement analyses using pose estimation have grown rapidly in the past few years, so this is a timely article that will be of interest to the field. My comments and suggestions are included below.

ABSTRACT

No comments.

INTRODUCTION

In the paragraph spanning lines 64-72, the authors introduce some issues with using pose estimation for gait analysis (e.g., false detection of multiple people in the image). However, previous studies (including the cited study by Stenum et al) have offered solutions for some of these issues. Given that the authors’ primary goal in this study is to focus on unconstrained environments, I suggest that they identify specifically the issues that remain with respect to unconstrained environments in this paragraph.

It also seems to me that the issues that this study aims to address are not specific to unconstrained environments. These issues may be present in unconstrained environments, of course, but they may also be present in some well-controlled environments as well. I suggest that the authors could emphasize that the analysis workflow that they present could be helpful for pose estimation-based gait analysis more broadly and not specifically in unconstrained environments.

Line 85 – I believe that the authors mean “frames per second”

There is a previous article that describes many of the issues with using currently available pose estimation algorithms for movement science (https://arxiv.org/pdf/1907.10226.pdf). It would be helpful if the authors could indicate in the introduction how their proposed study differs from this previous work or helps to resolve some of the issues proposed.

RESULTS

It is clear that the four types of anomalies identified are indeed problems for pose estimation-based gait analysis, but it is not clear how these four types of anomalies were selected. Presumably, there are other remaining issues – why/how were these four chosen in particular?

If I am understanding the results correctly, the “fix” that the authors propose for the anomalous errors is to essentially remove the data and perform gap filling, which is a rather crude solution. The authors provide some simulation work to address the reproducibility of the workflow, but this does not provide information about how accurately the gap-filled keypoints approximate ground-truth data. Comparison to some kind of ground-truth data (e.g., motion capture) would be helpful if available.

DISCUSSION

Similar to my comment about the introduction, it would be helpful if the authors could contextualize their approach and findings against many of the issues proposed in the Seethapathi et al paper. It should also be mentioned as a significant limitation that we do not yet understand the accuracy of this proposed approach.

Reviewer #2: The study evaluates the OpenPose tracker in terms of four types of anomalies to investigate whether the tracker errors are due to biological factors or technical errors. Removing identified technical errors, the tracker outputs can be used to perform gait analysis. The paper identifies anomalies and examines the tracker output on a large-scale gait dataset. The core idea of the evaluation looks valuable to understand if motion-capture or similar systems can be replaced with real-time pose trackers with less afford.

However I have two main concerns. 1) The paper categorizes anomaly types, but I am not sure if the proposed categorization truly identifies the actual types. 2) The authors presents a good simulation setup. However, the robustness of their post-processing technique can be better evaluated in terms of accuracy improvement using some ground truth data coming from a motion capture system. In the current version of the text, experimental results presented in Fig.4. looks like such an evaluation, where the ground truth comparison was used. But the experiment and its purpose are not clear with lack of details. The authors should improve the text and explain this part better.

Some comments:

The authors identify 10 different anomalies. They can improve their discussion on why and how these anomalies identified with related works from the literature. Otherwise proposed categorization could be subjective. For instance, undetected-parts (Fig1) or extreme lengthening are identified under anatomical constraints. However, it is not clear why it is not categorized as one category of the estimation error. Similarly, samples identified under COG category can be also identified under anatomical constraints. Therefore, authors should better present how they define these categories and support with related studies. Lines 116-196 presents the constraint but it is not clear what is the base for identifying biomechanics or physical constraints. This should be supported with reference works. Otherwise, it is not clear why we need the proposed categorization but not the estimation error.

The Result section can be revised. The relations of subsection are getting confused. For instance, in the current version of the text, it is not explicitly written that the sections in Line 116-168 expresses different categories of constraints to detect anomalies. Otherwise, it looks like anomalies are categorize as anatomical/biomechanical etc.. More text can help: Paragraph in lines 98-103 can be improved with few sentences or a new paragraph can be added before line 97 to express the purpose and relation of the following sections.

Some anomalies due to tracker error can be pruned based on self-evaluation of the video data, e.g. analysis on frames or consecutive frames to detect abnormal detections. Proposing such a model for post processing can be more robust than identifying various constraints with various threshold values. Therefore, the authors can add a discussion to present the advantages of their threshold-based model.

The accuracies in Fig4 can be given shortly in the text with more discussion on improvements. Assuming multiple anomaly types, the authors can explain in more detail how they compute the accuracies and how they correct the values. Moreover, what is the ground truth used in this experiment?

Line 374 - Line 419: In these sections, authors use some constant values. Can they give some reference study on these values? Are these values coming from some standards? Otherwise, the dataset contains various genders and age groups.

Line 389: The text can be revised to clearly identify parts related to ROM, COG and ankle distance. What part is related to ROM within the text?

Line 393-398: Please check the abbreviations. Some includes Rh, some others include RH (similarly RK and Rk).

Eq5: Can you please check the equation. This looks inconsistent with Eq.6

Eq7: Can you explain in the text what R_it is?

Eq8: Please check the equation. Does the third component need square?

The qualities of some figures are not good to view, higher resolution would be better.

**Have the authors made all data and (if applicable) computational code underlying the findings in their manuscript fully available?**

Reviewer #1: Yes

Reviewer #2: Yes

PLOS authors have the option to publish the peer review history of their article (what does this mean?). If published, this will include your full peer review and any attached files.

Reviewer #1: No

Reviewer #2: No
---

## [Decision Letter · Decision Letter 1]

21 Dec 2022

Dear Matsui,

We are pleased to inform you that your manuscript 'Types of anomalies in two-dimensional video-based gait analysis in uncontrolled environments' has been provisionally accepted for publication in PLOS Computational Biology.

Best regards,

Thurmon Lockhart

Guest Editor

PLOS Computational Biology

Feilim Mac Gabhann

Editor-in-Chief

PLOS Computational Biology

Many thanks for your nice paper.

Reviewer's Responses to Questions

**Comments to the Authors:**

Reviewer #1: The authors have adequately addressed my prior comments and suggestions. I thank them for sharing an interesting paper.

**Have the authors made all data and (if applicable) computational code underlying the findings in their manuscript fully available?**

Reviewer #1: Yes

PLOS authors have the option to publish the peer review history of their article (what does this mean?). If published, this will include your full peer review and any attached files.

Reviewer #1: No

---

## [Editor Report · Acceptance letter]

3 Jan 2023

PCOMPBIOL-D-22-00335R1 

Types of anomalies in two-dimensional video-based gait analysis in uncontrolled environments

Dear Dr Matsui,

I am pleased to inform you that your manuscript has been formally accepted for publication in PLOS Computational Biology. Your manuscript is now with our production department and you will be notified of the publication date in due course.

With kind regards,

Zsofia Freund
